# Association of Preoperative Copeptin Levels with Risk of All-Cause Mortality in a Prospective Cohort of Adult Cardiac Surgery Patients

**DOI:** 10.3390/cells13141197

**Published:** 2024-07-15

**Authors:** Mark G. Filipovic, Markus Huber, Beatrice Kobel, Corina Bello, Anja Levis, Lukas Andereggen, Ryota Kakizaki, Frank Stüber, Lorenz Räber, Markus M. Luedi

**Affiliations:** 1Department of Anaesthesiology and Pain Medicine, Inselspital, Bern University Hospital, University of Bern, 3010 Bern, Switzerland; markus.huber@insel.ch (M.H.); corina.bello@insel.ch (C.B.); anja.levis@insel.ch (A.L.); frank.stueber@insel.ch (F.S.); markus.luedi@extern.insel.ch (M.M.L.); 2Department for BioMedical Research (DBMR), University of Bern, 3010 Bern, Switzerland; beatrice.kobel@insel.ch; 3Department of Neurosurgery, Kantonsspital Aarau, 5001 Aarau, Switzerland; lukas.andereggen@ksa.ch; 4Faculty of Medicine, University of Bern, 3010 Bern, Switzerland; 5Department of Cardiology, Bern University Hospital, University of Bern, 3010 Bern, Switzerland; ryota.kakizaki@extern.insel.ch (R.K.); lorenz.raeber@insel.ch (L.R.); 6Department of Anaesthesiology and Pain Medicine, Cantonal Hospital of St. Gallen, 9000 St. Gallen, Switzerland

**Keywords:** outcome, inflammation, cardiac surgery, copeptin, cardiovascular

## Abstract

We aimed to investigate the association of preoperative copeptin, a new cardiovascular biomarker, with short- and long-term mortality in a cohort of adult patients undergoing cardiac surgery, including its potential as a prognostic marker for clinical outcome. Preoperative blood samples of the Bern Perioperative Biobank, a prospective cohort of adults undergoing cardiac surgery during 2019, were analyzed. The primary and secondary outcome measures were 30-day and 1-year all-cause mortality. Optimal copeptin thresholds were calculated with the Youden Index. Associations of copeptin levels with the two outcomes were examined with multivariable logistic regression models; their discriminatory capacity was assessed with the area under the receiver operating characteristic (AUROC). A total of 519 patients (78.4% male, median age 67 y (IQR: 60–73 y)) were included, with a median preoperative copeptin level of 7.6 pmol/L (IQR: 4.7–13.2 pmol/L). We identified an optimal threshold of 15.9 pmol/l (95%-CI: 7.7 to 46.5 pmol/L) for 30-day mortality and 15.9 pmol/L (95%-CI: 9.0 to 21.3 pmol/L) for 1-year all-cause mortality. Regression models featured an AUROC of 0.79 (95%-CI: 0.56 to 0.95) for adjusted log-transformed preoperative copeptin for 30-day mortality and an AUROC of 0.76 (95%-CI: 0.64 to 0.88) for 1-year mortality. In patients undergoing cardiac surgery, the baseline levels of copeptin emerged as a strong marker for 1-year all-cause death. Preoperative copeptin levels might possibly identify patients at risk for a complicated, long-term postoperative course, and therefore requiring a more rigorous postoperative observation and follow-up.

## 1. Introduction

Predicting clinical outcome after cardiac surgery is challenging, as mostly multimorbid patients undergo major surgery and postoperative performance is influenced by numerous and complexly interrelated perioperative factors [1]. While preoperative risk scores such as the European System for Cardiac Operative Risk Evaluation (EuroSCORE) II [2] are widely used, increasing efforts are being made to refine and individualize these scores with biomarkers to enable evidence-based and patient-centered decision-making before high-risk surgery [3,4,5].

Biomarkers of preoperative stress and inflammation have been reported as an associated factor with worse clinical outcome in cardiac and non-cardiac surgery [6]. Recently, the rapidly released peptide copeptin has been introduced as a new diagnostic and prognostic biomarker in cardiovascular disease [7]. Copeptin is co-released with arginine vasopressin (AVP) [8] in the hypothalamic stress response and is elevated in conditions such as sepsis, myocardial infarction, or stroke [9]. In contrast to AVP, it has improved diagnostic properties, including a longer half-life and better stability, making it an easy-to-measure surrogate marker [7]. In the perioperative setting, preoperative copeptin levels have been associated with higher rates of myocardial injury after non-cardiac surgery [10,11,12]. It was also reported that elevated postoperative copeptin levels further predicted acute kidney injury in cardiac surgery [13]. Notably, copeptin showed a tendency to predict the clinical outcome after congenital heart surgery in pediatric patients [14]. However, the association of preoperative copeptin levels with short- and long-term outcomes after cardiac surgery in adults remains elusive.

Therefore, we aimed to investigate the association between preoperative copeptin levels and short- and long-term mortality after cardiac surgery with cardiopulmonary bypass in adult patients and evaluate the predictive capacity of preoperative copeptin. Further, we wanted to compare the performance with EuroSCORE II [2] as a guideline-endorsed predictive risk model for short-term mortality. The benefit of both preoperative copeptin and EuroSCORE II for clinical decision-making is investigated by means of a decision curve analysis [15].

## 2. Materials and Methods

### 2.1. Study Population and Design

The present study was approved by the local ethics committee (Cantonal Ethics Commission of Bern, Bern, CH–KEK Nr. 2018-01272 for sampling and KEK Nr. 2019–2000 for data analysis). Written informed consent was obtained from every participant. Data were tested against a predefined hypothesis. The present study adhered to STROBE guidelines [16].

This study was performed as a retrospective analysis of the Bern Perioperative Biobank (ClinicalTrials.gov; NCT04767685). The Bern Perioperative Biobank is a prospective observational database with a sample of 519 adult patients who underwent cardiac surgery at the Bern University Hospital between January and December 2019. Inclusion criteria was scheduled elective cardiac surgery, while emergency surgeries and women with suspected or confirmed pregnancy were excluded. Cardiac surgeries performed included routine coronary artery bypass grafting (CABG), replacement or repair of aortic (AVR), mitral (MVR), and tricuspid (TVR) valves, and surgery of the ascending aorta or aortic arch. All patients received a median sternotomy and cardiopulmonary bypass, either with conventional extracorporeal circulation circuits (CECC) or minimally invasive extracorporeal circulation circuits (MIECC) [17].

### 2.2. Blood Sampling and Assessment of Copeptin

Blood samples (EDTA, ethylenediaminetetraacetic acid) were collected before the induction of general anesthesia (preoperative) and stored at −80 °C at the Bern Liquid Biobank. Copeptin levels were analyzed at the Center for Laboratory Medicine, Bern University Hospital, Inselspital Bern, according to standardized routine laboratory methods (Copeptin proAVP KRYPTOR by B·R·A·H·M·S GmbH, Hennigsdorf, Germany).

### 2.3. Outcome Measures and Other Study Variables

Thirty-day all-cause mortality was defined as the primary outcome measure and one-year all-cause mortality served as the secondary outcome measure.

All relevant pre-, peri-, and postoperative data (patient demographics, surgical, procedural, and anesthesiological data) for each patient were collected from electronic patient charts (Dendrite Clinical Systems Ltd., Henley-on-Thames, UK). Information on all-cause mortality was obtained from internal hospital records or from the national death registry. EuroSCORE II was calculated to assess the presumed risk of 30-day all-cause mortality.

### 2.4. Statistical Analysis

In terms of descriptive statistics, categorical variables were summarized by means of counts and frequencies. Numerical variables were summarized by the mean and standard deviation in the case of normally distributed variables and by the median and interquartile range (IQR) otherwise. For exploratory purposes only, unadjusted group comparisons of baseline values are shown in Table 1 and are based on the chi-square test for categorical variables and the t-test and unpaired two-sample Wilcoxon test for numerical variables.

We calculated optimal copeptin threshold levels with respect to 30-day and 1-year all-cause mortality based on the Youden Index [18]. Based on the optimal thresholds, patients were grouped into a high and low copeptin group. Kaplan–Meier plots for the two groups were used to show the corresponding survival probabilities over the course of the 1-year follow-up which were compared by means of a log-rank test. Hazard ratios adjusted for age, gender, and body mass index were calculated with a Cox proportional-hazards model.

The association of the logarithmic preoperative biomarker levels to the primary and secondary outcome measures were computed by means of multivariable Firth’s bias-reduced penalized-likelihood logistic regression with added covariate method [19], thus accounting for the low event rate of the outcomes. We adjusted for age, gender, and body mass index (BMI). Given that the magnitude of preoperative biomarker levels spanned several orders of magnitude, the logarithmic (base 10) preoperative levels were used as the covariate in the regression models.

The discrimination capacity of the optimal copeptin thresholds and the logistic regression models were assessed with the area under the receiver operating characteristic (AUROC). Calibration of the regressions models was examined with calibration curves [20].

The value of preoperative copeptin levels for decision-making in the context of mortality prediction was investigated with a decision curve analysis (DCA) [15]. Decision curves were computed for a suite of Firth’s bias-reduced penalized-likelihood logistic regression models and the associated net benefit was shown within a predefined range of treatment thresholds (0–10% probability range for 30-day all-cause mortality and 0–15% for 1-year all-cause mortality, respectively). Note that the treatment threshold refers to the minimum probability of a disease or an event at which a clinician would act, for example, running additional tests or a change of the treatment plan and allows representing different risk preferences by the clinicians (and patients) with respect to the clinical action of interest. Further information regarding the interpretation of decision curves can be found in the literature [21]. The uncertainty of the decision curves was quantified with 1000 bootstrap replicates and the median and 95%-confidence intervals of the net benefit for each treatment threshold were shown.

### 2.5. Missing Data and Statistical Software

Data availability of each variable is indicated in the corresponding table. There were two missing preoperative copeptin levels (n = 517). No imputation was performed for missing data; thus, each part of this study is based on a complete-case analysis. A *p* value < 0.05 was considered statistically significant. All computations were performed with R version 4.0.2 [22].

## 3. Results

Baseline characteristics of demographics and comorbidities, as well as surgical and procedural baseline characteristics stratified according to copeptin levels, are shown in Table 1. The median preoperative copeptin level was 7.6 pmol/L (IQR: 4.7–13.2 pmol/L). A scatterplot of copeptin vs. EuroSCORE II shows that the two quantities are significantly associated: patients with high preoperative copeptin levels show a high EuroSCORE II on average (*p* < 0.001) (Appendix A).

**Table 1 cells-13-01197-t001:** Baseline population, surgical, and procedural characteristics stratified according to preoperative copeptin.

	All Patients	Copeptin ≤15.9 pmol/L	Copeptin >15.9 pmol/L	*p*	n
	n = 519	n = 421 (81.1%)	n = 96 (18.9%)		
*Preoperative copeptin levels*					
**Copeptin**, pmol/L [IQR]	7.6 [4.7;13.2]	6.3 [4.2;9.2]	22.3 [18.4;32.3]	<0.001	517
*Demographics*					
**Age**, years [IQR]	67.0 [59.5;73.0]	67.0 [59.0;73.0]	69.0 [60.8;74.2]	0.15	519
**Sex** (Female), n (%)	112 (21.6%)	98 (23.3%)	14 (14.6%)	0.084	519
**Body Mass Index** (BMI; kg/m^2^), n [IQR]	26.1 [23.8;29.9]	25.9 [23.8;29.4]	27.6 [23.5;31.7]	0.088	519
*Comorbidities*					
**Diabetes**, n (%)	112 (21.6%)	80 (19.0%)	31 (32.3%)	0.006	519
**Hypertension**, n (%)	362 (70.3%)	283 (67.7%)	78 (82.1%)	0.008	515
**Dyslipidemia**, n (%)	309 (59.9%)	244 (58.2%)	63 (66.3%)	0.18	516
**Smoker**, n (%)				0.008	509
Non-smoker	250 (49.1%)	215 (52.1%)	34 (36.2%)		
Previous/current smoker	259 (50.9%)	198 (47.9%)	60 (63.8%)		
**Obesity** (BMI > 30 kg/m^2^), n (%)	124 (23.9%)	91 (21.6%)	32 (33.3%)	0.021	519
**Preoperative renal disease**, n (%)	96 (18.5%)	59 (14.0%)	37 (38.5%)	<0.001	519
**Peripheral vascular disease**, n (%)	34 (7.10%)	26 (6.70%)	8 (8.99%)	0.60	479
**Carotid disease**, n (%)	17 (3.89%)	15 (4.21%)	2 (2.53%)	0.75	437
**Myocardial infarction**, n (%)	57 (11.0%)	44 (10.5%)	13 (13.5%)	0.50	518
**COPD**, n (%)	49 (9.5%)	37 (8.87%)	12 (12.5%)	0.37	515
**NYHA** (>I), n (%)	325 (65.0%)	263 (64.8%)	61 (65.6%)	0.98	500
**CCS** (>0), n (%)	170 (34.6%)	141 (35.1%)	28 (31.5%)	0.60	492
**Ejection fraction**, % [IQR]	60 [55;65]	60 [55;65]	60 [45;60]	<0.001	512
**EuroSCORE II**, % [IQR]	1.8 [1.1;3.3]	1.7 [1.0;2.8]	2.7 [1.6;6.8]	<0.001	510
*Medications*					
**Betablocker**, n (%)	247 (47.6%)	185 (43.9%)	61 (63.5%)	0.001	519
**ACE**, n (%)	172 (33.1%)	134 (31.8%)	38 (39.6%)	0.18	519
**ARB**, n (%)	153 (29.5%)	110 (26.1%)	42 (43.8%)	0.001	519
**Aspirin**, n (%)	264 (50.9%)	221 (52.5%)	41 (42.7%)	0.11	519
**Statins**, n (%)	296 (57.0%)	236 (56.1%)	58 (60.4%)	0.51	519
**Steroids**, n (%)	19 (3.7%)	13 (3.1%)	6 (6.3%)	0.14	519
*Surgical/Procedural Characteristics*					
**ECC order MiECC**, n (%)				0.90	519
ECC	416 (80.2%)	337 (80.0%)	78 (81.2%)		
MiECC	103 (19.8%)	84 (20.0%)	18 (18.8%)		
**Operation duration**, min [IQR]	247 [206;297]	246 [202;290]	266 [221;318]	0.028	519
**Bypass time**, min [IQR]	116 [91.0;148]	111 [88;145]	132 [102;161]	0.001	519
**Aortic cross clamping**, min [IQR]	75.0 [59.0;102]	73 [57;99]	87 [63;114]	0.006	519
**Lowest body temperature**, C [IQR]	32.3 [31.7;33.6]	32.4 [31.7;33.7]	32.2 [31.7;33.4]	0.16	519
**Deep hypothermic cardiac arrest**, n (%)	49 (9.5%)	40 (9.5%)	8 (8.3%)	0.87	518
**Aortic valve**, n (%)	247 (47.6%)	204 (48.5%)	42 (43.8%)	0.47	519
**Mitral Valve**, n (%)	121 (23.3%)	93 (22.1%)	28 (29.2%)	0.18	519
**Tricuspid valve**, n (%)	46 (8.9%)	32 (7.6%)	14 (14.6%)	0.049	519
**Coronary artery bypass**, n (%)	221 (42.6%)	182 (43.2%)	38 (39.6%)	0.60	519
**Ascending aortic**, n (%)	113 (21.8%)	96 (22.8%)	16 (16.7%)	0.24	519
**Aortic arch**, n (%)	39 (7.5%)	31 (7.4%)	7 (7.3%)	>0.99	519

Data expressed as median [IQR] or number (%). ACE = Angiotensin Converting Enzyme Inhibitors, ARB = Angiotensin Receptor Blockers, BMI = Body Mass Index, CCS = Canadian Cardiovascular Society, COPD = Chronic Obstructive Pulmonary Disease, ECC = Extracorporeal Circulation Circuits, EuroSCORE = European System for Cardiac Operative Risk Evaluation, MiECC = Minimally Invasive Extracorporeal Circulation Circuits, NYHA = New York Heart Association.

Overall, 30-day all-cause mortality was 7/519 (1.4%, 95%-CI: 0.5% to 2.8%), while 1-year all-cause mortality was 16/519 (3.1%, 95%-CI: 1.8% to 5.0%), as displayed in Table 2. The corresponding preoperative copeptin levels for both groups are presented in Table 2 and graphically displayed in Figure 1A.

Multivariable logistic regression models revealed a significant adjusted association of log-transformed preoperative copeptin levels with 1-year all-cause mortality (OR 8.9, 95%-CI: 2.6 to 31.0, *p* < 0.001) but not 30-day all-cause mortality (OR 4.4, 95%-CI: 0.7 to 24.3, *p* = 0.11) (Table 3).

Based on the Youden Index, we identified an optimal threshold of 15.9 pmol/L (95%-CI: 7.7 to 46.5 pmol/L) for 30-day all-cause mortality and 15.9 pmol/L (95%-CI: 9.0 to 21.3 pmol/L) for 1-year all-cause mortality, respectively (Table 3 and Figure 1B). The predicted risk for 30-day all-cause mortality at the threshold was 1.7% (95%-CI: 0.5–3.2%), while it was 4.2% (95%-CI: 2.3–6.4%) for 1-year all-cause mortality (Figure 1B).

The regression models featured an AUROC of 0.79 (95%-CI: 0.56 to 0.95) for adjusted preoperative copeptin for 30-day all-cause mortality and an AUROC of 0.76 (95%-CI: 0.64 to 0.88) for 1-year all-cause mortality, respectively (Table 3). Table 3 further shows the associations of EuroSCORE II with the two mortality outcomes.

Based on the optimal threshold for preoperative copeptin, Figure 2 illustrates the survival curves for low and high copeptin groups, with a corresponding age–sex–BMI adjusted hazard ratio of 9.5 (95%-CI: 3.3 to 27.6, *p* < 0.001). An additional figure including the overall survival curve can be found in Appendix A.

Calibration curves and a decision curve analysis are displayed in Figure 3. While the regression models for 30-day all-cause mortality are well calibrated, the models for the outcome 1-year all-cause mortality show a degree of bias for high-risk patients, where the mortality risk is generally overestimated. Considering only preoperative copeptin levels provides little benefit for decision-making based on the prediction of 30-day all-cause mortality, where EuroSCORE II provides the largest benefit, in particular, for risk-averse settings (e.g., low threshold probabilities). Copeptin levels provide a large benefit for 1-year all-cause mortality decision-making. For threshold probabilities ranging from 3% to 10% (thus, less risk-averse settings), Figure 3D provides evidence that the largest benefit for decision-making results from combining EuroSCORE II and copeptin levels (solid green line).

## 4. Discussion

We found an association of preoperative copeptin levels with long-term (1-year) but not short-term (30-day) all-cause mortality in adults undergoing cardiac surgery. With respect to these two outcomes, we identified an optimal threshold of 15.9 pmol/L in this cohort. Based on regression models and a decision curve analysis, as well as in comparison to EuroSCORE II, preoperative copeptin levels might serve as a potential biomarker for risk prediction for long-term outcome after cardiac surgery with cardiopulmonary bypass.

Our findings are consistent with a growing body of research, establishing copeptin as an emerging diagnostic and prognostic biomarker in cardiac disease [7,23,24]. Copeptin derives from the precursor protein pre-provasopressin together with AVP, the main regulating hormone of body fluid homeostasis [25]. Copeptin is a 39-amino-acid-long glycosylated peptide with a leucine-rich core region and a relatively low molecular weight of approximately 5 kDa [25]. While AVP plays a pivotal role in the endocrine stress response, the physiological function of copeptin remains largely unknown [26]. Previous works have suggested that copeptin is a chaperone-like molecule for pro-AVP in the structural formation of the AVP precursor [27,28]. This function seems to mediate through an interaction with the calnexin/calreticulin system, which monitors protein folding; however, no specific copeptin receptors or elimination mechanisms are known [29].

Copeptin has been studied as a clinical indicator for diagnosis and for prognosis in various diseases [9]. At baseline, patients above the copeptin threshold were more likely to have traditional cardiovascular risk factors such as diabetes and hypertension. Similarly, a lower median ejection fraction and a higher median EuroSCORE II in the above-threshold group support the role of copeptin as a marker of cardiovascular disease, as described below. In addition, kidney disease was more common in the higher copeptin group. Several studies have shown that copeptin plays an important role in the diagnosis, prognosis, and possibly even the pathogenesis of chronic kidney disease [30]. However, the complex relationship between circulating levels of copeptin and impaired renal function remains the subject of scientific debate [30].

Copeptin has been extensively studied in acute coronary syndromes [31,32,33] and was proposed to rule out myocardial infarction in addition to troponin measurements [34,35]. In heart failure, its properties as a predictive marker for incidence and outcome have been repeatedly demonstrated [36,37,38]. Concerning cardiac surgery, in a small cohort (n = 20), Homs et al. showed that individuals with normal preoperative copeptin levels and a rapid postoperative return to baseline were likely to have an uneventful postoperative recovery [39]. Additionally, in a study of pediatric patients scheduled for congenital heart surgery, a tendency to predict clinical outcome by measuring copeptin levels at baseline was identified [14]. Further, a Polish sample of patients undergoing coronary artery bypass grafting showed a trend towards increased 30-day mortality with higher copeptin levels [40]. Contrarily, preoperative copeptin levels in our cohort were not associated with short-term but moreover with long-term mortality.

Concerning the predictive capacities of preoperative copeptin, we identified an optimal threshold of 15.9 pmol/l for both the prediction of 30-day and 1-year all-cause mortality. Multivariable logistic regression models adjusted for age, sex, and BMI yielded a medium AUROC for the prediction of 30-day all-cause mortality (0.76) and for the prediction of 1-year all-cause mortality (0.79).

For short-term mortality prediction, our data suggest that there is no added benefit of isolated preoperative copeptin values compared to EuroSCORE II for decision-making and only minimal benefit in combining the two. Both preoperative copeptin and EuroSCORE II identify patients with a higher overall risk, as patients with high preoperative copeptin at baseline on average also score higher with EuroSCORE II. Nevertheless, EuroSCORE II seems superior in short-term risk prediction, which is not further astonishing, as we are comparing a simple, isolated biomarker to a complex, established predictive tool specifically designed for this scenario [2].

For 1-year all-cause mortality, however, preoperative copeptin levels provide a large predictive benefit according to our data. Copeptin potentially reflects the overall cardiovascular functional status of an individual, including subtle pathological conditions which remain undiagnosed at the time of surgery but gain importance in the determination of long-term outcomes. Therefore, copeptin might additionally identify individuals who need a more rigorous postoperative observation and follow-up. For other cardiovascular disease such as acute coronary syndromes, the “2020 European Society of Cardiology Guidelines for the management of acute coronary syndromes in patients presenting without persistent ST-segment elevation” [41] recommend against the measurement of copeptin for routine risk or prognosis assessment. However, copeptin might have a potential stand in preoperative risk assessment for long-term outcomes before cardiac surgery. Limitations of our study include the single-center design preventing not including external validation. Further, statistical analysis was adjusted for the potential confounders age, gender, and BMI. However, this study is limited in controlling for other potential confounders due to the limited sample size and low event rates. For example, preoperative kidney disease was more frequent in the group with the higher copeptin levels, potentially confounding our results. However, the relationship between circulating levels of copeptin and renal function is still the subject of scientific debate [30]. Additionally, we were limited to preoperative copeptin levels and have not described the perioperative dynamics, which might be influenced by factors such as CPB time, aortic cross clamping, and deep hypothermic cardiac arrest. Cardiac surgery and the use of CPB are known to induce a unique physiological response and parallel dynamics in biomarkers, which in turn are associated with outcomes [42,43,44]. Potential future studies should focus on these issues and include a larger sample size to comprehensively identify further confounders and test the robustness of our findings.

## 5. Conclusions

In conclusion, copeptin is a potential candidate for individual risk assessment prior to cardiac surgery in adults. Preoperative copeptin levels might detect patients at risk for a complicated, long-term postoperative course. These patients require a more rigorous postoperative observation and follow-up. Copeptin should be further investigated in terms of its predictive capabilities for outcomes after cardiac surgery with cardiopulmonary bypass in order to allow improved individual risk assessment and management.

## Figures and Tables

**Figure 1 cells-13-01197-f001:**
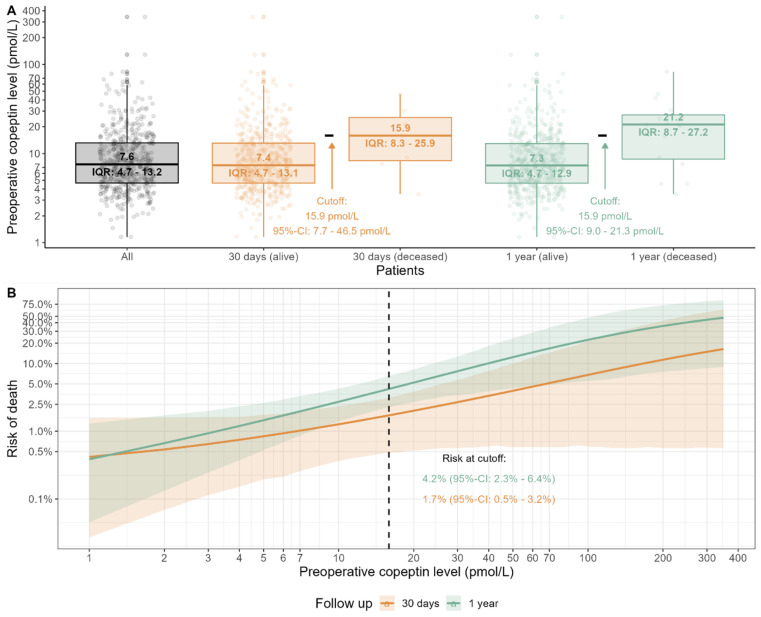
(**A**) Scatterplot of preoperative copeptin levels and matched ranges for all patients, both alive and deceased, after 30 days and 1 year, respectively. (**B**) Predicted risk of death, both for 30-day and 1-year follow-up, by a logistic regression model featuring log-10 transformed preoperative copeptin levels as covariate. Mean (solid lines) and 95%-confidence intervals (ribbons) are shown. The predicted risk at the optimal copeptin cutoff values is also shown. IQR = Interquartile Range.

**Figure 2 cells-13-01197-f002:**
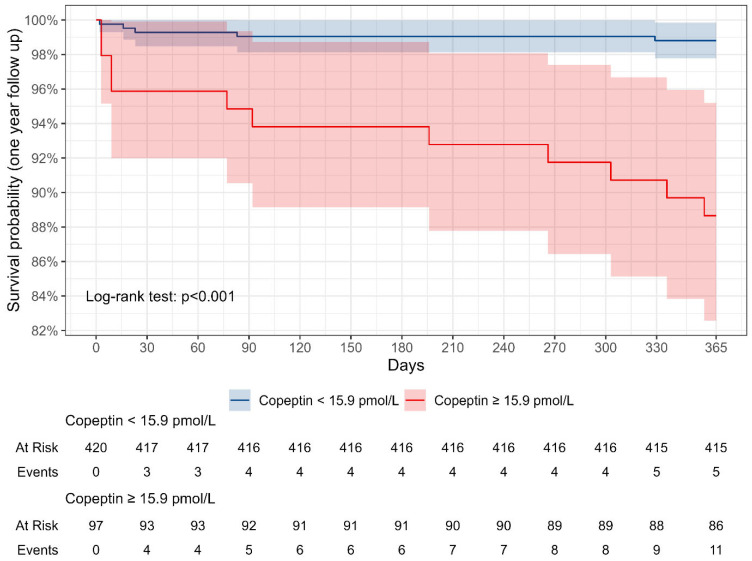
Kaplan–Meier curve for 1-year follow-up. The survival probability was stratified according to low (<15.9 pmol/L, blue line) and high (≥15.9 pmol/L, red line) preoperative copeptin levels.

**Figure 3 cells-13-01197-f003:**
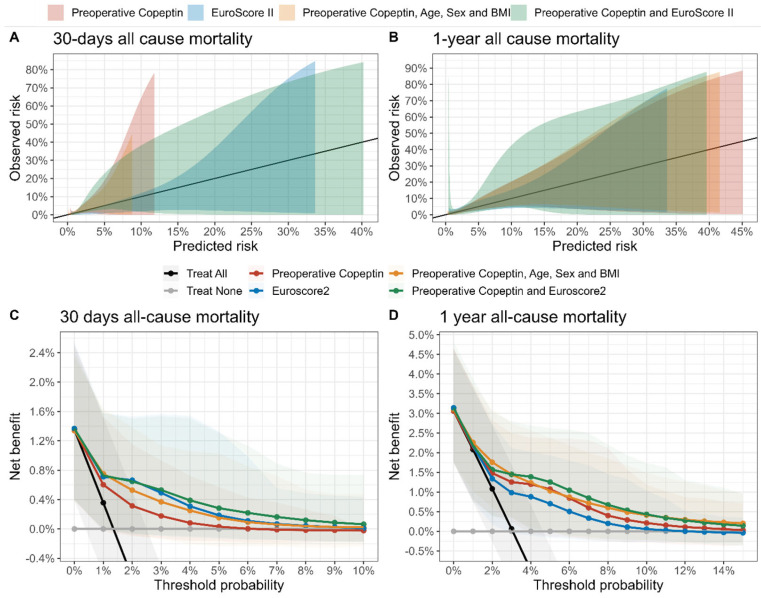
Calibration curves and decision curve analysis (DCA) for a suite of uni- and multivariable logistic regression models with either 30-day or 1-year mortality as the outcome. Optimal calibration is indicated in panels (**A**,**B**) as solid black lines. The mean and associated 95%-confidence intervals for the net benefit within the predefined range of treatment threshold probabilities in the DCA are shown (**C**,**D**).

**Table 2 cells-13-01197-t002:** Thirty-day and one-year all-cause mortality and corresponding ranges in copeptin levels.

Mortality Outcome	Survived	Deceased
**30-day all-cause mortality**	512/519 (98.7%, 95%-CI: 97.2% to 99.5%)	7/519 (1.4%, 95%-CI: 0.5% to 2.8%)
**1-year all-cause mortality**	503/519 (96.9%, 95%-CI: 95.0% to 98.2%)	16/519 (3.1%, 95%-CI: 1.8% to 5.0%)
**Copeptin**	Survived	Deceased
**30-day**	7.4 pmol/L (IQR: 4.7 to 13.1 pmol/L)	15.9 pmol/L (IQR: 8.3 to 25.9 pmol/L)
**1-year**	7.3 pmol/L (IQR: 4.7 to 12.9 pmol/L)	21.2 pmol/L (IQR: 8.7 to 27.2 pmol/L)

**Table 3 cells-13-01197-t003:** Association of preoperative copeptin levels with outcome measures, optimal thresholds, and modelling results based on Firth’s logistic regression with added covariate method. AUROC = Area Under the Receiver Operating Characteristic.

*Optimal thresholds based on Youden Index*	**30-day all-cause mortality**	**1-year all-cause mortality**
**Preoperative copeptin levels** (pmol/L)	15.9 pmol/L (95%-CI: 7.7 to 46.5 pmol/L)	15.9 pmol/L (95%-CI: 9.0 to 21.3 pmol/L)
*AUROC*	0.69 (95%-CI: 0.42 to 0.90)	0.74 (95%-CI: 0.59 to 0.89)
**Euroscore II**	9.1 (95%-CI: 1.8 to 10.7)	5.0 (95%-CI: 1.1 to 9.2)
*AUROC*	0.80 (95%-CI: 0.57 to 0.97)	0.69 (95%-CI: 0.54 to 0.83)
*Multivariable logistic regression models (odds ratios)*	**30-day all-cause mortality**	**1-year all-cause mortality**
**Model 1:**		
**Preoperative copeptin levels (pmol/L) (log10)**	4.4 (95%-CI: 0.7 to 24.3, *p* = 0.11)	8.9 (95%-CI: 2.6 to 31.0, *p* < 0.001)
*AUROC*	0.69 (95%-CI: 0.46 to 0.90)	0.74 (95%-CI: 0.59 to 0.88)
**Model 2:**		
**Preoperative copeptin levels** (pmol/L) (log10)	3.2 (95%-CI: 0.5 to 18.0, *p* = 0.22)	7.8 (95%-CI: 2.2 to 27.4, *p* = 0.002)
**Age** (years)	1.07 (95%-CI: 0.99 to 1.19, *p* = 0.09)	1.03 (95%-CI: 0.98 to 1.09, *p* = 0.25)
**Sex** (Male)	1.4 (95%-CI: 0.3 to 13.8, *p* = 0.71)	1.5 (95%-CI: 0.4 to 7.8, *p* = 0.57)
**Body Mass Index** (BMI; kg/m^2^)	1.04 (95%-CI: 0.91 to 1.17, *p* = 0.53)	0.95 (95%-CI: 0.86 to 1.05, *p* = 0.34)
*AUROC*	0.79 (95%-CI: 0.56 to 0.95)	0.76 (95%-CI: 0.64 to 0.88)
**Model 3:**		
**Preoperative copeptin levels** (pmol/L) (log10)	2.6 (95%-CI: 0.3 to 46.6, *p* = 0.36)	6.8 (95%-CI: 1.9 to 24.5, *p* = 0.004)
**Euroscore II**	1.12 (95%-CI: 1.00 to 1.24, *p* = 0.04)	1.07 (95%-CI: 0.98 to 1.16, *p* = 0.13)
*AUROC*	0.77 (95%-CI: 0.46 to 0.97)	0.76 (95%-CI: 0.61 to 0.89)

## Data Availability

Data availability is restricted to protect confidential information. Data will be made available upon request, with permission for the purposes of peer review. Clinical trial number and registry: ClinicalTrials.gov (NCT04767685).

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
