# Peer review of "Association of Preoperative Copeptin Levels with Risk of All-Cause Mortality in a Prospective Cohort of Adult Cardiac Surgery Patients"

_cells, 2024, doi:10.3390/cells13141197_

Round 1
Reviewer 1 Report
Comments and Suggestions for Authors
It was a pleasure to review this manuscript. It is well written and easy to be followed.
The study group have a significant amount of subjects. The statistical analysis is correct and the results well presented.
I would the authors to add further information regarding copeptin. Are there any factors that effect the results? For example age, BMI, gender?
Otherwise, the discussion is very well written and the authors discussed satisfactory the future value of their results.
Author Response
Comment 1: It was a pleasure to review this manuscript. It is well written and easy to be followed. The study group have a significant amount of subjects. The statistical analysis is correct and the results well presented.
Response 1: Thank you very much for taking the time to review this manuscript and your positive initial evaluation. Please find the detailed responses below and the corresponding revisions/corrections highlighted/in track changes in the re-submitted files.
Comment 2: I would the authors to add further information regarding copeptin. Are there any factors that effect the results? For example age, BMI, gender? Otherwise, the discussion is very well written and the authors discussed satisfactory the future value of their results.
Response 2: Thank you for raising this very important issue. Our models correct for age, BMI and gender, so this should not be a source of potential bias. A distribution of both patient and surgical characteristics stratified according to higher and lower copeptin levels can be found in Table 1. These findings suggest that the burden of cardiovascular disease was higher in the group with higher copeptin levels, in line with our current understand of copeptin and the findings of this study. Referring also to the second reviewer’s comment, we have additionally commented on the higher prevalence of renal disease in the group above the copeptin threshold. Altogether, we have extended the discussion section and commented on potential limitations.
Reviewer 2 Report
Comments and Suggestions for Authors
Filipovic and colleagues present an interesting work on the prognostic potential of copeptin following cardiac surgery. I would like to congratulate the authors on the well written manuscript. I have the following comments:
- Copeptin is partly eliminated renally, there was a higher number of patients with chronic kidney disease in the Copeptin> 15.9 pmol/L group. (Christ-Crain et. al 2016)
- Did comorbidities affect the levels of copeptin ? (Abdelmageed et al. 2023)
- There was a significant difference in the duration of CPB and aortic cross clamping. Could this affect the release of copeptin ?
- Deep hypothemic cardiac arrest should be excluded form this cohort.
- The AUC values reported are under 0.8, which suggest that copeptin is a mediocre marker at best. The authors should consider a larger sample size.
Author Response
Comment 1: Filipovic and colleagues present an interesting work on the prognostic potential of copeptin following cardiac surgery. I would like to congratulate the authors on the well written manuscript. I have the following comments:
Response 1: Thank you very much for taking the time to review this manuscript and your positive initial evaluation. Please find the detailed responses below and the corresponding revisions/corrections highlighted/in track changes in the re-submitted files. We hope to have sufficiently addressed your valued comments and believe that they have significantly contributed to improving the quality of our manuscript.
Comment 2: Copeptin is partly eliminated renally, there was a higher number of patients with chronic kidney disease in the Copeptin> 15.9 pmol/L group. (Christ-Crain et. al 2016)
Response 2: Thank you for raising this very important issue. As you pointed out, copeptin has been shown to play an important role in the diagnosis, prognosis and potentially pathogenesis of chronic renal disease (Igleasis et al. 2022). However, the relationship between circulating levels of copeptin and impaired renal function are complex and subject to scientific debate and cannot be attributed to a renal excretion alone. For example, copeptin was not associated with GFR in living kidney donors after nephrectomy despite a significant fall in renal function (Zittema et al. 2014). This suggests that GFR alone is not a major determinant of copeptin and vice versa.
However, we cannot exclude that renal function might bias our results. Therefore we have included a comment in the discussion section and extended the limitations section according to this reviewer’s important advice.
Comment 3: Did comorbidities affect the levels of copeptin ? (Abdelmageed et al. 2023)
Response 3: Thank you for this important question. Our models correct for age, BMI and gender, so this should not be a source of potential bias. A distribution of both patient and surgical characteristics stratified according to higher and lower copeptin levels can be found in Table 1. These findings suggest that the burden of cardiovascular disease was higher in the group with higher copeptin levels, in line with our current understand of copeptin and the findings of this study. We have extended the discussion section accordingly.
Comments 4: There was a significant difference in the duration of CPB and aortic cross clamping. Could this affect the release of copeptin ?
Response 4: Thank you for raising this important issue. One could imagine that both factors might influence the release of copeptin. However, to the best of our knowledge, there are no previous studies investigating this specific question. Further, our study investigates the preoperative copeptin levels, which are therefore not influenced by perioperative events. We have added this limitation to the according section.
Comment 5: Deep hypothemic cardiac arrest should be excluded form this cohort.
Response 5: Thank you for this important comment. In line with the comment above, our study investigates the preoperative copeptin levels, which are therefore not influenced by perioperative events. We acknowledge that this is a true limitation and have therefore included the following statement in the discussion section: “Additionally, we were limited to preoperative copeptin levels and have not described the perioperative dynamics, which might be influenced by factors such as ECC time, aortic cross clamping and deep hypothermic cardiac arrest. Potential future studies should focus on these issues and include a larger sample size to comprehensively identify further confounders and test the robustness of our findings.”
Comment 6: The AUC values reported are under 0.8, which suggest that copeptin is a mediocre marker at best. The authors should consider a larger sample size.
Response 6: Thank you, we totally agree with this reviewer’s comment and have included an according statement in the limitations section as elaborated above. Despite the limited sample size and the identified limitations, we believe that our findings are innovative and might lay ground for a comprehensive evaluation of copeptin in the context of cardiovascular surgery.
Response 4: Thank you for raising this important issue. One could imagine that both factors might influence the release of copeptin. However, to the best of our knowledge, there are no previous studies investigating this specific question. Further, our study investigates the preoperative copeptin levels, which are therefore not influenced by perioperative events. We have added this limitation to the according section.
Comments 5: Deep hypothemic cardiac arrest should be excluded form this cohort.
Thank you for this important comment. In line with the comment above, our study investigates the preoperative copeptin levels, which are therefore not influenced by perioperative events. We acknowledge that this is a true limitation and have therefore included the following statement in the discussion section: “Additionally, we were limited to preoperative copeptin levels and have not described the perioperative dynamics, which might be influenced by factors such as ECC time, aortic cross clamping and deep hypothermic cardiac arrest. Potential future studies should focus on these issues and include a larger sample size to comprehensively identify further confounders and test the robustness of our findings.”
Comments 6: The AUC values reported are under 0.8, which suggest that copeptin is a mediocre marker at best. The authors should consider a larger sample size.
Thank you, we totally agree with this reviewer’s comment and have included an according statement in the limitations section as elaborated above. Despite the limited sample size and the identified limitations, we believe that our findings are innovative and might lay ground for a comprehensive evaluation of copeptin in the context of cardiovascular surgery.